# ADVERSARIAL REINFORCEMENT LEARNING FRAMEWORK FOR ESP CHEATER SIMULATION

## ABSTRACT

Extra-Sensory Perception (ESP) cheats, which reveal hidden in-game information such as enemy locations, are difficult to detect because their effects are not directly observable in player behavior. The lack of observable evidence makes it difficult to collect reliably labeled data, which is essential for training effective anti-cheat systems. Furthermore, cheaters often adapt their behavior by limiting or disguising their cheat usage, which further complicates detection and detector development. To address these challenges, we propose a simulation framework for controlled modeling of ESP cheaters, non-cheaters, and trajectory-based detectors. We model cheaters and non-cheaters as reinforcement learning agents with different levels of observability, while detectors classify their behavioral trajectories. Next, we formulate the interaction between the cheater and the detector as an adversarial game, allowing both players to co-adapt over time. To reflect realistic cheater strategies, we introduce a structured cheater model that dynamically switches between cheating and non-cheating behaviors based on detection risk. Experiments demonstrate that our framework successfully simulates adaptive cheater behaviors that strategically balance reward optimization and detection evasion. This work provides a controllable and extensible platform for studying adaptive cheating behaviors and developing effective cheat detectors.

## 1 INTRODUCTION

Cheating is a persistent issue in digital games that significantly violates the fairness and degrades the user experience. If cheating is not properly addressed, it may drive players away from the game, ultimately leading to reduced revenue for game developers. It has been reported to cause an estimated loss of $29 billion and to drive away 78% of gamers for a year (Irdeto, 2022).

One of the most representative forms of cheating is Extra-Sensory Perception (ESP), which allows cheaters to access hidden game information. A common example is the use of wallhacks in first-person shooter (FPS) games: while normal players cannot see opponents behind walls, cheaters can, enabling them to avoid unexpected attacks and easily eliminate enemies. These unfair advantages severely disrupt the balance of the game, especially in settings where players must make decisions and formulate strategies based on incomplete information. ESP cheats are notoriously hard to prevent due to their passive nature: they do not modify game files or client-side data. In addition, their usage is often indistinguishable from normal gameplay when observed from a third-person perspective, making them especially challenging to detect through in-game monitoring or user reports (PUBG: BATTLEGROUNDS Anti-Cheat Team, 2024).

Another major challenge in detecting ESP cheats is the lack of a reliable dataset for developing anti-cheat systems. First, it is hard to clearly distinguish between cheaters and non-cheaters. Cheaters may adopt various strategies: some use cheats consistently throughout a game, some toggle them on and off, while others behave as if they are not cheating - taking suboptimal actions or mimicking normal gameplay - to avoid suspicion. Moreover, they may also pretend their advantageous actions are coincidences in order to appear as normal players. Because of these strategies, even with monitoring, it is difficult to judge whether a player is actually cheating or just getting lucky. Sometimes, non-cheaters with unusually good luck are falsely flagged as cheaters and punished. Even when penalties are imposed, it is very rare for players to admit that they used cheats. As a result, a significant number of cases can be mislabeled.

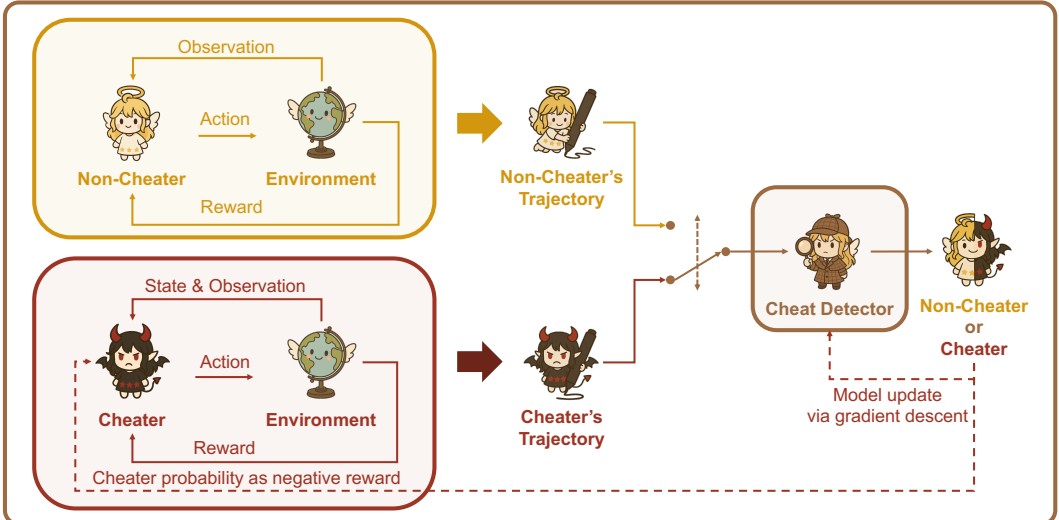

Figure 1: Overview of ESP cheater simulation framework. The cheat detector discriminates the trajectory whether it was generated by a non-cheater or a cheater. After detector making its decision, the cheater updates its policy based on the detection result in order to evade detection. Simultaneously, the cheat detector updates itself to improve its classification accuracy.

Next, cheaters continuously evolve their strategies to evade detection (Tao et al., 2018; Jonnalagadda et al., 2021). When a cheat detector is developed and deployed, cheaters adapt by limiting their cheat usage just enough to avoid detection. For example, if a simple threshold-based detection algorithm is used, cheaters will intentionally operate just below the threshold to remain undetected. As a result, the statistical characteristics of past cheater data differ from those of recent data. Since only recent data is useful for developing effective detectors, it becomes difficult to collect a large amount of high-quality training and test data.

Due to these challenges, constructing a reliable real-world dataset is difficult. It hinders the development and evaluation of cheat detectors that are both trustworthy and high-performing. To address these limitations, we adopt simulation as an alternative approach. Simulation offers several key advantages. First, it provides full control over the environment, allowing us to assign ground-truth labels with complete certainty. Second, simulation can be scaled as needed, allowing us to generate a large amount of training data to support robust detector development. Last, simulation enables us to reproduce and study various cheating strategies mentioned earlier, providing a controlled way to train detectors against diverse and sophisticated behaviors.

To this end, we introduce the ESP simulation framework that simulates cheaters, non-cheaters, and cheat detectors with their co-evolving behaviors. In the partially observable environment, we model the cheater and the non-cheater as reinforcement learning (RL) agents with different levels of observability. We use a classifier as a cheat detector to estimate the probability of a trajectory being generated by the cheater. We formulate the interaction between players as a minimax problem, enabling both to co-evolve during training. An overview of the framework is illustrated in Fig. 1.

To verify the effectiveness of the framework, we implement *Gridworld* and *Blackjack* environments that can simulate both fully observable and partially observable agents. These simplified single-agent environments provide interpretable behavioral indicators, such as exploration patterns and trajectory length, which allow us to quantitatively and qualitatively analyze the behaviors of cheaters under controlled conditions. Although these environments are abstracted, they capture fundamental aspects such as spatial exploration and reward-driven behavior that also appear in more complex game settings. In particular, even in FPS games, player movements and interactions can often be represented as grid-like trajectories on a minimap, making the Gridworld environment a meaningful abstraction for studying such dynamics. While our long-term goal is to extend the framework to complex multi-player environments such as FPS games, these experiments serve as a foundational baseline toward understanding the fundamental co-adaptation mechanism between cheaters and de-

tectors, before extending the analysis to complex multi-agent environments. We also utilize various detector designs based on the trajectory, trajectory length, and reward. With these diverse configurations, we conduct extensive experiments that demonstrate the effectiveness of our framework in modeling adaptive cheaters. Our code is available at link.

## 2 RELATED WORKS

**Cheat Detection for Games.**   Several studies (Alayed et al., 2013; Tao et al., 2018; Jonnalagadda et al., 2021; Pinto et al., 2021; Kanervisto et al., 2022; Zhang et al., 2024) have explored cheat detection in video games, proposing a variety of approaches for identifying different types of malicious behaviors. Alayed et al. (2013) used a support vector machine (SVM) (Suthaharan, 2016) to detect aimbots based on server-side logs from online FPS games. Tao et al. (2018) proposed NGUARD, a bot detection framework for multiplayer online role-playing games (MMORPGs), which combines supervised and unsupervised learning to detect known bots and discover previously unseen ones. Jonnalagadda et al. (2021) introduced a vision-based deep neural network (DNN) detector to identify illicit on-screen overlays in *Counter-Strike: Global Offensive* (CS:GO) (Valve, 2012). In addition, they applied the interval bound propagation method (Gowal et al., 2018) to defend against adversarial attacks targeting the detection system. Pinto et al. (2021) formulated sequences of keyboard and mouse events as a multivariate time series, and employed a convolutional neural network (CNN) (O'shea & Nash, 2015) to detect triggerbots and aimbots in CS:GO. Kanervisto et al. (2022) developed an aimbot that mimics human behavior using a generative adversarial network (GAN) (Goodfellow et al., 2020), and evaluated the performance of multiple detection methods against it. Zhang et al. (2024) proposed HAWK, an anti-cheat framework for CS:GO, which leverages machine learning to mimic the decision process of human experts in detecting wallhacks and aimbots. Despite recent advances, most approaches assume that cheaters behave in fixed, non-adaptive ways. It overlooks the fact that real-world cheaters often adjust their strategies to avoid detection, highlighting the need for simulation frameworks that can model such adaptive behaviors.

**Adversarial Training in Reinforcement Learning.**   Adversarial learning traces its roots to the GAN (Goodfellow et al., 2020), where a generator and a discriminator engage in a minimax game to match data distributions. Ho & Ermon (2016) later carried this game-theoretic structure over to reinforcement learning with *generative adversarial imitation learning* (GAIL), which trains a policy to fool a discriminator that separates expert and learner trajectories, thereby reproducing the expert's state–action occupancy without an explicit reward-recovery step. For *robustness*, Pinto et al. (2017) proposed *robust adversarial reinforcement learning* (RARL), where an adversary injects disturbance forces into the simulator so that the resulting policy remains stable under variations in friction, mass, and other model mismatches. Zhang et al. (2020) proposed *state-adversarial Markov decision process* (SA-MDP) framework to analyze the observation robustness of RL algorithms and introduced a policy regularization technique to enhance their robustness. Sun et al. (2022) established a theoretical understanding of the optimality of adversarial attacks from the perspective of policy perturbations. Zhang et al. (2021) proposed *alternating training with learned adversaries* (ATLA), a framework that improves an agent's robustness under perturbed observations by jointly training the adversary and the RL agent. Franzmeyer et al. (2024) introduced the *illusory attack*, an information-theoretic adversarial framework that ensures statistically grounded detectability while maintaining attack effectiveness against both human and AI agents. Dennis et al. (2020) presented *protagonist–antagonist induced regret environment design* (PAIRED), training an environment designer and an antagonist demonstrator to build a curriculum of tasks that are solvable yet currently unsolved by the protagonist, thereby boosting zero-shot transfer. The adversarial paradigm has even reached the language domain: the *self-playing adversarial language game* (ALG) (Cheng et al., 2025) places two large language models in a hide-and-seek game with taboo tokens, iteratively sharpening their reasoning strategies across benchmarks. Although these lines of work pursue diverse goals in imitation (Ho & Ermon, 2016), robustness (Pinto et al., 2017; Zhang et al., 2020; 2021; Sun et al., 2022; Franzmeyer et al., 2024) to observation or policy perturbations, curriculum-driven generalization (Dennis et al., 2020), and domain expansion (Cheng et al., 2025), they all leverage adversarial interaction to enhance policy capability. In contrast, our framework jointly trains an ESP cheater and a trajectory-level detector in a partially observable setting where the detector provides the detection probability as a negative reward. This design captures the dynamic evolution of cheating behaviors that balance reward optimization and detection evasion.

Although the overall formulation of this work is similar to the minimax structure of GAN or GAIL, the modeling purpose is fundamentally different. GAN and GAIL aim to imitate the data distribution or policies, focusing on designing highly expressive generators and improving distribution-matching performance. In contrast, the cheater in our framework does not necessarily imitate the non-cheater's policy. It instead learns new behaviors that balance the trade-off between detection avoidance and reward maximization. While imitation may help evade detection, it cannot achieve the cheater's primary goal, which is obtaining unfair advantages through higher rewards. Another key difference of our study is that we treat the detector's performance as an equally important subject of analysis. In conventional GAN or GAIL frameworks, the discriminator serves merely as an auxiliary tool for generator training, and its own performance is rarely analyzed. In contrast, our framework considers the detector as an active player. This approach reflects the unique characteristics of the cheat detection domain and provides a clear conceptual distinction from GAN and GAIL.

## 3 ESP SIMULATION FRAMEWORK

We model the game environment as a partially observable Markov decision process (POMDP) (Åström, 1965; Kaelbling et al., 1998), defined by the tuple $\langle S, A, T, r, \Omega, O, \gamma \rangle$, where $S$ is the set of environment states, $A$ is the set of actions, $T$ denotes the state transition probability distribution, $r : S \times A \to \mathbb{R}$ is the reward function, $\Omega$ is the set of observations available to the non-cheater player, $O$ is the observation function, defining the conditional observation probabilities, and $\gamma \in [0, 1]$ is a discount factor. Consistent with previous works related to the trajectory feedback (Liu et al., 2019; Efroni et al., 2021), we assume $\gamma = 1$ throughout this paper.

We consider three different players in this game: the ESP cheater, the non-cheater, and the cheat detector. To simplify the problem, we make the following assumptions:

1. There is only a single cheater, a single non-cheater, and a single cheat detector.

2. Given a trajectory, a sequence of states and actions $\tau = (s_0, a_0, s_1, a_1, \cdots, s_t, a_t)$, the detector estimates the probability that it was generated by the cheater. During the inference, the detector receives no information about the ground-truth label of the trajectory. On the other hand, it is allowed to access the ground-truth label during the training.

3. The non-cheater operates under partial observability. On the other hand, the ESP cheater has full observability of the environment and can directly accesses the state. The cheater also has access to the observations available to the non-cheater, as well as the cheater probability assigned by the detector to its generated trajectory.

4. Players are bounded rational (Simon, 1955; Kahneman & Tversky, 1982; Selten, 1990). According to the concept of the bounded rationality, decision makers often struggle to find a global optimum due to their limited information, time, and computational resources. As a result, they remain at a nearby local optimum. Experimental studies (Nagel, 1995; Coricelli & Nagel, 2009) also support this claim by showing that players in games do not act rationally and their behavior is bounded. From a game theory perspective (Flåm, 1998; Chen et al., 2011; Ratliff et al., 2016), this implies that players tend to reach the local Nash equilibrium (Alos-Ferrer & Ania, 2001) and rarely shift to the different local optimal strategy.

Under these assumptions, both the cheater and the non-cheater are using local optimal policies. It can produce a large performance gap between the two players, making it easier for the detector to distinguish them than in real-world scenarios. In other words, this setup represents an upper bound on the detector's performance.

We now define the following components: the non-cheater's policy $\pi_n : \Omega \times A \to [0, 1]$, the cheater's policy $\pi_c : S \times \Omega \times A \to [0, 1]$, and the cheat detector's classifier $D : \mathcal{T} \to [0, 1]$, where $\mathcal{T}$ denotes the set of all possible trajectories. Since the non-cheater follows the local optimal policy under partial observability, $\pi_n$ can be obtained by solving a reward maximization problem $\max_{\pi_n} J(\pi_n)$ with policy optimization algorithms such as A2C (Mnih et al., 2016) and PPO (Schulman et al., 2017), where $J(\pi_n)$ denotes the expected return under policy $\pi_n$. Note that it is guaranteed to reach the local optimum when using temporal difference (Maei et al., 2009), deep Q-learning (Fan et al., 2020), and actor-critic methods (Holzleitner et al., 2021; Tian et al., 2023).

We then formulate the dynamic interaction between the ESP cheater and the detector as an adversarial game. In this setup, the detector aims to improve its classification performance by analyzing behavior patterns exhibited by the agents, while the cheater continually adapts its strategy to evade detection and maximize in-game rewards. Formally, the detector updates its classifier $D$ to distinguish between trajectories induced by the cheater policy $\pi_c$ and the non-cheater policy $\pi_n$, by minimizing the binary cross-entropy loss over samples $\tau_c \sim \pi_c$ and $\tau_n \sim \pi_n$. Simultaneously, the cheater optimizes its policy $\pi_c$ to maximize the expected return $J(\pi_c)$ while minimizing $D(\tau_c)$, thereby reducing the classifier's confidence in identifying cheating behaviors. It is formalized as a minimax game:

$$\min_{\pi_c} \max_D \mathbb{E}_{\tau_c \sim \pi_c} \left[\log D(\tau_c)\right] + \mathbb{E}_{\tau_n \sim \pi_n} \left[\log(1 - D(\tau_n))\right] - \lambda^{-1} J(\pi_c), \tag{1}$$

where $\lambda \geq 0$ is an adversarial coefficient. It is a hyperparamter that controls the trade-off between maximizing in-game rewards and minimizing detector's detectability.

To find the local Nash equilibrium, we employ an alternating optimization approach with the gradient descent-ascent (GDA) algorithm, in which the cheater policy $\pi_c$ and the classifier $D$ are iteratively trained. Specifically, equation 1 can be decomposed into two sub-problems:

$$\text{Optimize cheater policy}: \quad \max_{\pi_c} J(\pi_c) - \lambda \mathbb{E}_{\tau_c \sim \pi_c} \left[\log D(\tau_c)\right] \tag{2}$$

$$\text{Optimize cheat detector}: \quad \max_D \mathbb{E}_{\tau_c \sim \pi_c} \left[\log D(\tau_c)\right] + \mathbb{E}_{\tau_n \sim \pi_n} \left[\log(1 - D(\tau_n))\right]. \tag{3}$$

Note that Daskalakis & Panageas (2018); Adolphs et al. (2019); Mazumdar et al. (2020) theoretically proved that applying GDA to a minimax problem yields a local Nash equilibrium.

In practice, $\log D$ in equation 2 may not provide sufficiently strong gradients for effective optimization of $\pi_c$, as discussed in Goodfellow et al. (2020). When the classifier becomes confident in its predictions, $\log D(\tau)$ tends to saturate, thereby diminishing its influence on the objective. To alleviate this issue, we adopt $-\log(1 - D(\tau))$ in place of $\log D(\tau)$, as suggested in Goodfellow et al. (2020). It yields the following surrogate objective:

$$\text{Optimize cheater policy (Practical)}: \quad \max_{\pi_c} J(\pi_c) + \lambda \mathbb{E}_{\tau_c \sim \pi_c} \left[\log(1 - D(\tau_c))\right]. \tag{4}$$

To solve the equation 4, we apply reward shaping (Ng et al., 1999) with the shaped reward $r'_t$:

$$r'_t = \begin{cases} r_t, & \text{if } t < |\tau_c| - 1 \\ r_t + \lambda \log(1 - D(\tau_c)), & \text{if } t = |\tau_c| - 1, \end{cases} \tag{5}$$

where $r_t$ denotes the original reward at timestep $t$. This transformation reformulates the objective in equation 4 into a standard expected return form $\mathbb{E}[\sum_t r'_t]$ as follow:

$$J(\pi_c) + \lambda \mathbb{E}_{\tau_c \sim \pi_c}[\log(1 - D(\tau_c))] = \mathbb{E}_{\tau_c \sim \pi_c}[\sum_t r_t + \lambda \log(1 - D(\tau_c))] = \mathbb{E}_{\tau_c \sim \pi_c}[\sum_t r'_t]. \tag{6}$$

It enables the use of standard policy optimization algorithms, such as A2C, PPO and others.

**Structured Cheater Modeling.** In actual gameplay, cheaters rarely invent new tactics. They tend to behave like non-cheaters most of the time and exploit cheating only at critical moments to gain a decisive advantage. In other words, a realistic cheater behaves as a mixture of a non-cheater and a pure cheater, which represents the local optimal agent under full observability without any constraints related to detectability.

To reflect this behavior, we model the cheater policy $\pi_c$ as an interpolation of the non-cheater policy $\pi_n$ and the pure cheater policy $\pi_c^{(p)} : \mathcal{S} \times \Omega \times \mathcal{A} \to [0, 1]$, inspired by the mixture-of-experts architecture (Jacobs et al., 1991; Shazeer et al., 2017):

$$\pi_c(a|s,o) = \mathcal{I}_{\omega(s,o)}[\pi_n(a|o), \pi_c^{(p)}(a|s,o)], \tag{7}$$

where $\omega : \mathcal{S} \times \Omega \to [0, 1]$ denotes the interpolation weight function. In this paper, we adopt a linear interpolation: $\pi_c(a|s,o) = (1 - \omega(s,o)) \cdot \pi_n(a|o) + \omega(s,o) \cdot \pi_c^{(p)}(a|s,o)$. Similar to the non-cheater policy, the pure cheater policy can be obtained by solving reward maximization problem

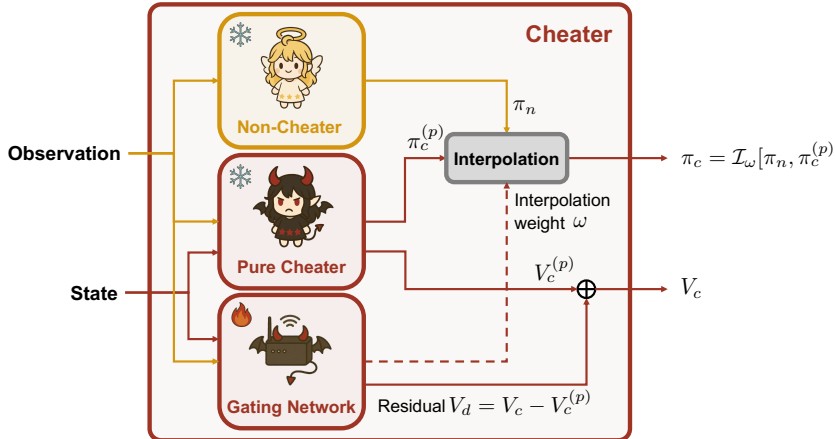

Figure 2: Actor-critic model architecture of the cheater. The model consists of three components: the non-cheater, the pure cheater, and the gating network. Only the gating network is trainable. The gating network produces an interpolation weight $\omega$ used to combine the two policies, as well as the residual value $V_d = V_c - V_c^{(p)}$ corresponding to the expected penalty by the detector.

---

**Algorithm 1** Adversarial training of cheater and cheat detector

---

**Input:** Non-cheater policy $\pi_n$, pure cheater policy $\pi_c^{(p)}$, initial cheat detector $D$ and adversarial coefficient $\lambda$
**Output:** Cheater policy $\pi_c$ and updated $D$
 Initialize interpolation weight function $\omega$
 Initialize $\pi_c \leftarrow \mathcal{I}_\omega[\pi_n, \pi_c^{(p)}] \cdots (7)$
 **for** iteration=$1, 2, \cdots$ **do**
  Sample trajectories $\tau_c^i \sim \pi_c$, $\tau_n^i \sim \pi_n$
  Optimize $\omega$ by maximizing expected return with reward shaping (5) using $\{\tau_c^i\}$
  Optimize $D$ by minimizing cross entropy (3) using $\{\tau_c^i\}$ and $\{\tau_n^i\}$
  Update $\pi_c \leftarrow \mathcal{I}_\omega[\pi_n, \pi_c^{(p)}] \cdots (7)$
 **end for**

---

$\max_{\pi_c^{(p)}} J(\pi_c^{(p)})$ with policy optimization algorithms. $\omega$ acts as a routing function, controlling the degree to which the agent behaves like a cheater. It is a learnable function, enabling the agent to adaptively decide when to cheat based on the current state and observation. It allows the cheater policy to interpolate smoothly between cheating and non-cheating behaviors based on context.

Next, we can use the value function $V_c^{(p)}$ of the pure cheater as an initial point to train the value function $V_c$ of the cheater. Specifically, we decompose the value function $V_c$ into two parts: $V_c^{(p)}$ estimates the return expected under the pure cheater policy, while the residual $V_d = V_c - V_c^{(p)}$ serves as an auxiliary component that accounts for the potential penalty associated with detection risk and the value difference induced by the mixture of policies. $V_d$ is modeled as a learnable function, allowing the agent to infer contextual detectability, account for the mixture of policies, and adjust its value estimate accordingly. We illustrate the detailed architecture of the cheater in Fig. 2. The full optimization process with the structured cheater modeling is provided in Algo. 1.

## 4 EXPERIMENTS

### 4.1 EXPERIMENT SETUP

**Environments.** To test the framework and the optimization algorithm described in Sec. 3, we design custom single-agent environments: *Gridworld* and *Blackjack*. Refer to Appendix A for the detailed explanations about environments. The Gridworld (Fig. 3a) is a two-dimensional grid world

| Dealer | Player | Deck | Action |
|--------|--------|------|--------|
| T$\diamondsuit$, (3$\clubsuit$) | 5$\diamondsuit$, 3$\diamondsuit$ | 6$\clubsuit$, (8$\spadesuit$), (2$\spadesuit$), (T$\spadesuit$), (9$\diamondsuit$), (7$\clubsuit$), (9$\clubsuit$), (7$\spadesuit$), (6$\heartsuit$), (A$\diamondsuit$), (6$\spadesuit$), (A$\clubsuit$), (A$\heartsuit$) | hit, stand |

(a) Gridworld               (b) Blackjack

Figure 3: (a) Visualization of the Gridworld environment. The figures show walls in gray, the agent as a red triangle, items as green circles, and lava as orange regions with black wave patterns. Only the $3 \times 3$ region in front of the agent is visible. (b) Visualization of the Blackjack environment. Cards with the parenthesis are invisible to the non-cheater.

| Game | Player type | AP | AUROC | Average reward | Average trajectory length |
|------|-------------|----|-------|----------------|---------------------------|
| Gridworld | Non-cheater | $0.500_{\pm 0.000}$ | $0.500_{\pm 0.000}$ | $4.676_{\pm 0.005}$ | $62.133_{\pm 1.762}$ |
| | Pure cheater | $0.772_{\pm 0.012}$ | $0.810_{\pm 0.011}$ | $4.759_{\pm 0.011}$ | $45.708_{\pm 2.471}$ |
| Blackjack | Non-cheater | $0.500_{\pm 0.000}$ | $0.500_{\pm 0.000}$ | $-0.031_{\pm 0.002}$ | $1.473_{\pm 0.141}$ |
| | Pure cheater | $0.798_{\pm 0.049}$ | $0.818_{\pm 0.021}$ | $0.704_{\pm 0.057}$ | $1.146_{\pm 0.027}$ |

Table 1: Performances of the pretrained agents.

with randomly distributed items, walls, and lava. The agent controls its orientation and position using three actions: *TurnLeft*, *TurnRight*, and *MoveForward*. By default, only a small square region in front of the agent is visible. On the other hand, the cheater agent can observe every grid. The agent receives a time-decaying reward when collecting the item and incurs a time-decaying penalty when encountering lava. The episode ends when the agent has collected all items or the maximum number of timesteps has elapsed. After the episode ends, the cheat detector receives the agent's trajectory as form of image and discriminates whether the agent is a cheater or not.

The Blackjack (Fig. 3b) is a card game that the player aims to hold cards whose count does not exceed 21 while exceeding the dealer's count. The player can use four actions to draw the card from the deck or to update the bet: *Hit*, *Stand*, *DoubleDown*, and *Surrender*. By default, the player can observe their hand (their initial hand + revealed cards from the deck) and the dealer's upcard. In contrast, the cheater can observe every card, including the dealer's hold card and the deck. After the episode ends, the player receives or loses the bet depending on their count. Then, the detector discriminates the cheater based on the trajectory including cards and the action history.

**Evaluation Metrics.** We employ three main metrics to evaluate the performances of the cheater and the detector: *Average Precision (AP)*, *Area Under the Receiver Operating Characteristic curve (AUROC)*, and *Average reward*. AP measures the area under the precision–recall curve, and AU-ROC measures the area under the ROC curve representing the trade-off between true and false positive rates. A higher AUROC reflects a stronger ability to identify cheaters and avoid false alarms. AP and AUROC evaluate the detector's ability to distinguish cheater trajectories from non-cheater ones, while the average reward measures the effectiveness of the policy in the game environment. Lower AP and AUROC indicate that the cheater successfully deceives the detector. In addition, we report *Average trajectory length* to provide further insights into the agent behavior. Under the non-adversarial setting, non-cheaters tend to produce longer trajectories than cheaters, since they must actively explore the environment to search for items or check the next card, leading to longer episode length. Lastly, we use *Relative reward* to visualize the reward changes over detectability. It is defined as $(\text{reward} - \text{non-cheater's reward})/(\text{pure cheater's reward} - \text{non-cheater's reward})$, to represent the normalized reward gain of the adversarial cheater compared to the pure cheater.

**Implementation Details.** We use a CNN-based actor-critic architecture for the policy network and a CNN-based classifier for the detector network. Both the cheater and non-cheater policies are trained using PPO algorithm (Schulman et al., 2017). Before conducting adversarial training, we construct the trajectory dataset for each policy and then pretrain the detector. The pretraining results are summarized in Tab. 1. For adversarial training, we finetune the cheater policy and the detector network based on Alg. 1. More details can be found in Appendix B.

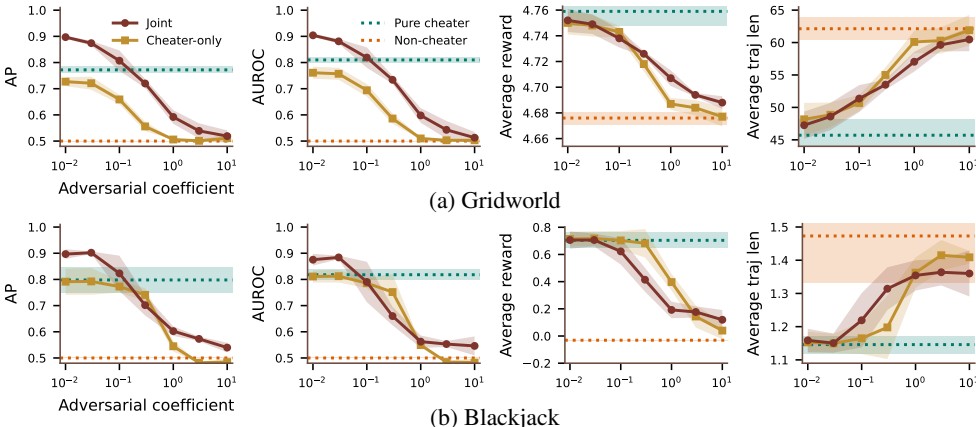

Figure 4: Performance metrics as functions of the adversarial coefficient $\lambda$. We plot the experimental results of two different settings: updating both the cheater and the detector (*Joint*) and updating only the cheater with the fixed detector (*Cheater-only*). As $\lambda$ increases, the cheater becomes harder to detect (lower AP and AUROC). The average reward decreases gradually, showing that the cheater sacrifices efficiency to avoid detection. Average trajectory length increases as the cheater takes longer and less efficient choices to appear less suspicious.

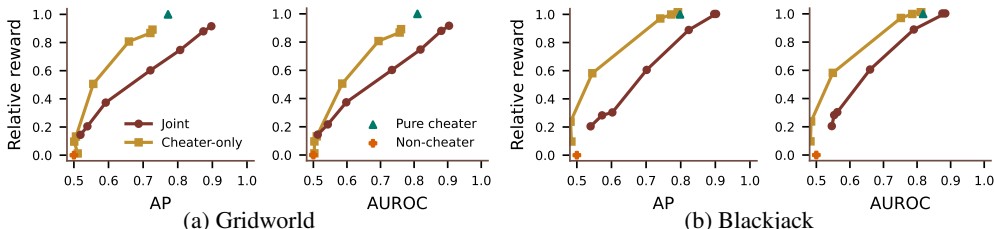

Figure 5: Reward changes over detectability. We can interpret figures from two different perspectives. (1) At an equivalent level of detectability across detectors, the cheater can get less reward with the adversarially trained detector compared to the fixed detector. (2) At an equivalent level of cheater reward, the adversarially trained detector achieves higher detectability than the fixed detector.

## 4.2 ADVERSARIAL TRAINING

To validate the effectiveness of our adversarial simulation framework, we conduct the experiment to analyze how the cheater adapts its behavior under different detection pressures. We vary the adversarial coefficient $\lambda$ in objective equation 4, which controls the trade-off between maximizing in-game rewards and minimizing the probability of being detected. The selected values for $\lambda$ are 0.01, 0.03, 0.1, 0.3, 1, 3, and 10.

Fig. 4 presents quantitative results in an equilibrium under different values of $\lambda$. As $\lambda$ increases, the cheater policy places more emphasis on deceiving the detector. The detector's ability to correctly identify cheater trajectories significantly decreases, as indicated by lower AP and AUROC. Alongside this, we observe a gradual decline in the cheater's average reward. For Gridworld, this trend is closely related to an increase in average trajectory length: As the cheater attempts to appear less suspicious, it avoids taking the most direct and efficient paths to collect items. Instead, it follows longer and less direct routes, similar to the exploratory behavior of non-cheaters, as illustrated in Fig. 6a. For Blackjack, the cheater reduces aggressive behaviors such as surrender or double down. Instead, it adopts more conservative actions that resemble the cautious playstyle of the non-cheater, as shown in Fig. 6b. While these behaviors help the cheater evade detection, they also delay item collection or reduce the bet, resulting in a lower reward.

Interestingly, for small values of $\lambda \leq 0.1$, the detector achieves even higher AP and AUROC than when trained against the pure cheater. During adversarial training, the cheater policy evolves gradually, generating a wide variety of trajectories as it explores different cheating behaviors. This

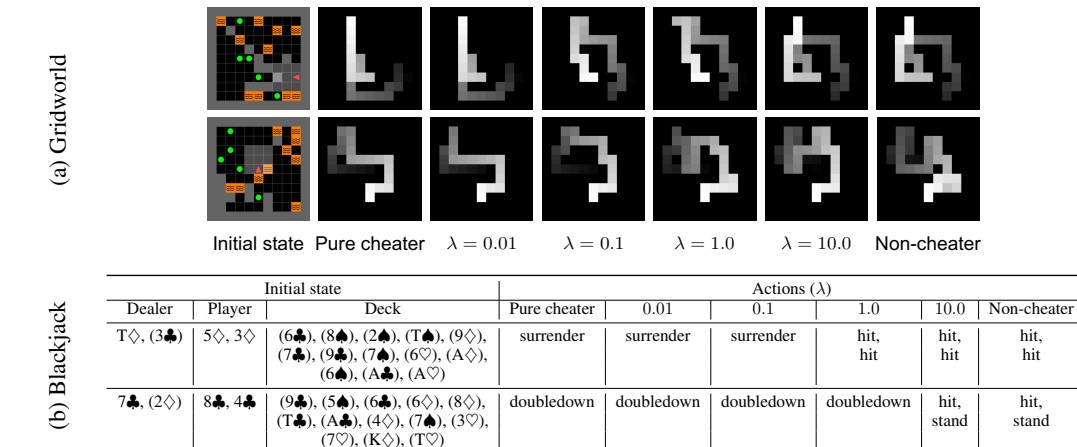

Figure 6: Trajectories generated by cheaters trained under different $\lambda$. As $\lambda$ increases, the cheater tends to avoid optimal behavior and takes less direct paths to stay hidden from the detector.

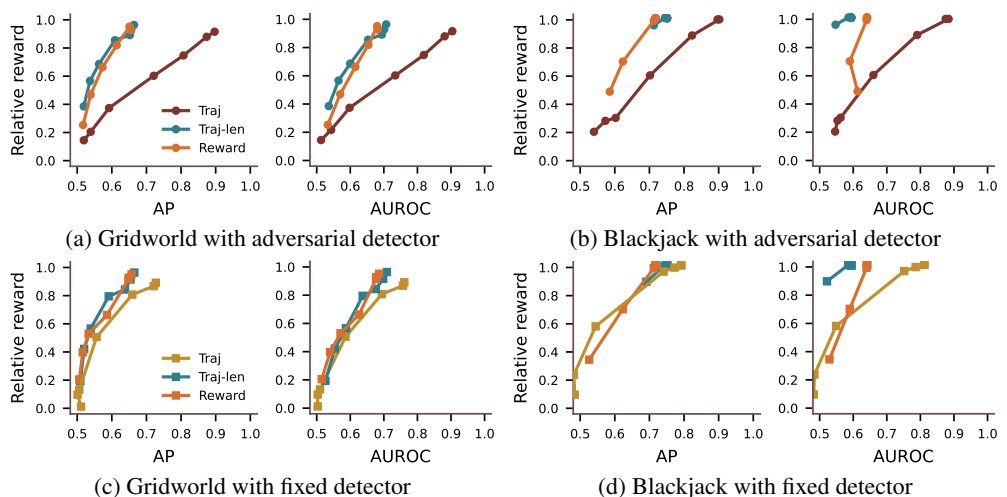

Figure 7: The trajectory-based detector (*Traj*) outperforms both the trajectory-length-based (*Traj-len*) and the reward-based detector (*Reward*) when using adversarially trained detectors. On the other hand, there is not much performance difference between detectors when using fixed detectors.

exposes the detector to a more diverse set of training examples compared to the static pure cheater case, allowing it to learn a more robust classifier. However, since the cheater in this case focuses mainly on reward maximization and does not actively attempt to avoid detection, it fails to deceive the improved detector. To support this observation, we present the performance of all combinations of pretrained and adversarially trained cheaters and detectors under $\lambda = 0.01$ in Tab. 3 in the Appendix. The adversarially trained detector outperforms the pretrained one across both AP and AUROC, regardless of the cheater's training method. This suggests that the detector becomes more robust than when trained solely against a pure cheater. For intermediate values $0.1 < \lambda < 1.0$, the cheater achieves better evasion than the pure cheater, but the detector still performs reasonably well, with AP and AUROC remaining above 0.6. In contrast, when $\lambda \geq 1.0$, both AP and AUROC drop to around 0.5-0.6, indicating that the detector can no longer reliably distinguish cheater trajectories from those of non-cheaters. It suggests that the cheater has learned highly deceptive behavior, effectively making the detector useless. Fig. 5 shows that the cheater still achieves a higher average reward than the non-cheater, despite having lower AP and AUROC. Specifically, the adversarially trained cheater retains approximately 30-40% of the reward advantage that the pure cheater has over the non-cheater at an AP and AUROC of 0.6. It demonstrates that the cheater can effectively evade detection while still consistently gaining an advantage over multiple episodes.

### 4.3 ABLATION STUDIES

To analyze the adaptability of the cheater, we conduct an additional experiment where the detector is kept fixed and only the cheater policy is updated. In Fig. 4, we observe that the overall trends in the average reward and the average trajectory length remain similar to those in the adversarial training setting. However, the overall detection performance is lower than in the adversarial training case. Even when $\lambda$ is very small ($\lambda = 0.01$), the detection scores remain below the pure cheater's scores. Since the detector does not update, the cheater can repeatedly reinforce its policy in the same direction, leading to a stable reduction in detection scores. As a result, the cheater is able to achieve higher rewards for the same detection score compared to the adversarial training setting, as shown in Fig. 5. It highlights the importance of continuously updating the detector, as a fixed detector may be insufficient to counter adaptive cheaters over time.

Next, we analyze how the design choice of the detector affects detection performance. In addition to the trajectory-based detector used in our main experiments, we use trajectory-length-based and reward-based detectors. Refer to Appendix D for details of detectors. As shown in Fig. 7, the trajectory-based detector consistently outperforms others when using adversarially trained detectors. It suggests that trajectory information is more effective in identifying cheating behavior than trajectory length or reward alone. In contrast, when using fixed detectors, the performance differences among the three designs are marginal. It highlights that without continuously adapting detectors against evolving cheaters, even a well-designed detector provides limited benefit.

Lastly, we investigate the effect of the structured cheater modeling architecture. We demonstrate that training without the structured modeling may lead to unstable training dynamics, especially when the adversarial pressure is high (i.e., adversarial coefficient $\lambda$ is large). We provide detailed descriptions and experimental results related to this in Appendix C.

## 5 CONCLUSIONS AND FUTURE WORKS

In this paper, we present the ESP simulation framework, which models ESP cheaters, non-cheaters, and cheat detectors with their adversarial relationships. Experimental results demonstrate that our framework can effectively simulate adaptive cheater behaviors that balance reward optimization and evade detection. We show that the fixed detector can be easily exploited by adaptive cheaters, emphasizing the need for continuously updated detection mechanisms. Although the setup is disadvantageous to the cheater, we further analyze that the cheater can outperform the non-cheater in terms of average reward, while remaining undetected by a trajectory-based detector. These results suggest that detecting adaptive cheaters may require more sophisticated approaches, such as utilizing behavior patterns observed across multiple episodes.

While this work focuses on simple game environments, we believe it can be extended to more complex settings. First, it would be valuable to experiment with games that involve more intricate rules and strategic decision-making, such as FPS games or multiplayer environments. Studying cheater behavior and detector performance in multiplayer settings would provide more realistic and practical insights for building robust anti-cheat systems. To achieve this, it is necessary to develop a more efficient learning methodology than the current GDA-based approach to shorten the time required for policy training. In this regard, the theoretical analysis of the efficiency, stability, and convergence characteristics of the optimization process would be an important research direction. Next, it would be valuable to check how similar the simulated players are to the real-world players. Regarding this, we have made strong assumptions that both the cheater and non-cheater are single locally optimal players, and the cheater has access to the cheater probability from the detector. In practical scenarios, however, multiple non-optimal players exist, and only binary signals are available from the detector. It could produce a gap between the real and the simulation. Therefore, we plan to extend the framework to include multiple cheaters and non-cheaters at different levels of play, with binary feedback from the detector for a more realistic simulation.

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

| Grid size | Agent's view size | # items | # walls | # lava | Maximum episode length |
|:---:|:---:|:---:|:---:|:---:|:---:|
| $11 \times 11$ | $3 \times 3$ | 5 | 10 | 10 | 484 |

Table 2: Default configuration of the Gridworld environment.

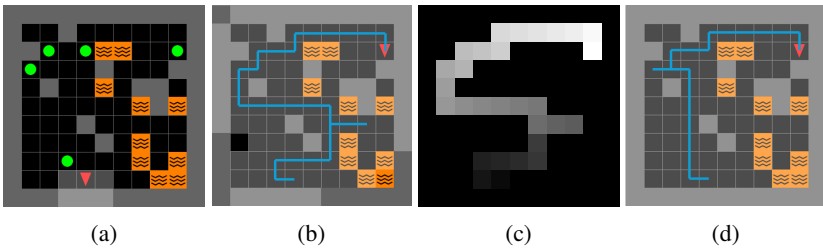

| (a) | (b) | (c) | (d) |

Figure 8: Visualizations of the Gridworld environment and the agent behavior. The figures show walls in gray, the agent as a red triangle, collectible items as green circles, and lava as orange regions with black wave patterns. (a) Initial state and observation: Only the $3 \times 3$ region in front of the agent is visible. (b) Retrospective board and the movement history (Non-cheater): Light-colored tiles represent areas that have been observed by the agent. The blue line traces the agent's history throughout the episode. (c) History heatmap: A heatmap of the agent's movement over time, with brighter colors indicating more recent positions. (d) Retrospective board and the movement history (Cheater): All tiles are visible, even for areas the agent has not approached.

## A  ENVIRONMENT DETAILS

### A.1  GRIDWORLD

**Environment Layout.**  Gridworld environment is a two-dimensional grid world based on the MiniGrid (Chevalier-Boisvert et al., 2023) framework. It contains $n \times n$ tiles, surrounded by walls that prevent the agent from leaving the environment. At the beginning of each episode, we randomly place the agent with the random facing direction. We also randomly distribute the collectible items, the walls, and the lava across empty cells. Fig. 8a shows the example of the initial state and the initial observation for the agents. The episode ends when the agent has collected all items or the maximum number of timesteps has elapsed.

The agent interacts with the environment through a discrete action space $\mathcal{A}$, consisting of three primitive actions: *TurnLeft*, *TurnRight*, and *MoveForward*. These actions control the agent's orientation and position within the grid. When the agent executes *MoveForward* and enters a tile containing an item, the item is immediately collected and removed from the environment. Upon collecting the item, the agent receives $1 - t/T$ as a reward, where $t$ is the current timestep and $T$ is the maximum length of the episode. This time-decaying reward encourages the agent to collect items as quickly as possible. When the agent steps into a lava tile, the lava is removed, and the agent receives a time-decaying penalty of $-0.1 \times (1 - t/T)$, discouraging such behavior. Tab. 2 summarizes the default environment configurations used throughout our experiments.

**Input Structures.**  The cheater and the non-cheater receive observation dictionaries consisting of five components: *Agent view*, *Movement history*, *Retrospective board*, *Item*, and *Time*. *Agent view* provides a top-down visual representation of the environment from the agent's perspective at the current timestep. Only a small square region in front of the agent is visible to them. *Movement history* is represented as a heatmap that records the agent's movement over time; each cell stores the timestep at which the agent last visited that location, with unvisited cells set to zero. We normalize the recorded timestep values to the range $[0, 1]$, where higher values correspond to more recent visits. Fig. 8c is an example of the movement history heatmap. *Retrospective board* (Gao et al., 2019) is a global memory map that records the latest observed information for each tile. Whenever a tile becomes visible to the agent, the retrospective board is updated accordingly. Non-cheater agents can only access their own partially constructed retrospective board based on their observations, whereas cheater agents have access to the full environment without any visibility restrictions. Fig. 8b and 8d

| | Initial Hand | Deck / Action | | Initial Hand | Deck / Action |
|---|---|---|---|---|---|
| Dealer | T$\diamondsuit$, (3$\clubsuit$) | (6$\clubsuit$), (8$\spadesuit$), (2$\spadesuit$), (T$\spadesuit$), (9$\diamondsuit$), (7$\clubsuit$), (9$\clubsuit$), (7$\spadesuit$), (6$\heartsuit$), (A$\diamondsuit$), (6$\spadesuit$), (A$\clubsuit$), (A$\heartsuit$) | Dealer | T$\diamondsuit$, (3$\clubsuit$) | 6$\clubsuit$, (8$\spadesuit$), (2$\spadesuit$), (T$\spadesuit$), (9$\diamondsuit$), (7$\clubsuit$), (9$\clubsuit$), (7$\spadesuit$), (6$\heartsuit$), (A$\diamondsuit$), (6$\spadesuit$), (A$\clubsuit$), (A$\heartsuit$) |
| Player | 5$\diamondsuit$, 3$\diamondsuit$ | | Player | 5$\diamondsuit$, 3$\diamondsuit$ | hit, stand |

(a) Initial state      (b) Final state

Figure 9: Visualizations of the Blackjack environment. Cards with the parenthesis are invisible to the non-cheater. (a) is the example initial state and (b) is its final state. Player can only see their hand (initial hand + revealed cards from the deck), dealer's upcard, and the player's action history.

illustrate examples of the retrospective board constructed by the non-cheater and the cheater agent. *Item* denotes the number of items collected by the agent, and *Time* indicates the current timestep within the episode.

For the cheat detector, we encode each trajectory as a two-channel image. The first channel represents the agent's movement history in the form of a heatmap. The second channel captures the initial state of the board, including items, walls, and lava.

## A.2    BLACKJACK

**Environment Layout.**    Blackjack is a famous card game played with standard playing cards. The player aims to hold cards whose count does not exceed 21 while exceeding the dealer's count. The game uses a standard 52-card deck. Card values are as follows: cards from 2 to 9 take their numeric values, 10, J, Q, and K count as 10, and A counts as either 1 or 11.

Before the game starts, the player places a unit bet of 1. The player and the dealer then receive two cards each. The dealer reveals one upcard and keeps one hold card hidden. For each turn, the player can choose one of the actions: *Hit*, *Stand*, *DoubleDown*, and *Surrender*. *Hit* draws one additional card. *Stand* stops drawing card. *DoubleDown* doubles the bet, takes exactly one more card, and then stands. *Surrender* loses half of the bet and finishes the game immediately. To ensure that it is only available at the first turn, the player loses immediately and forfeits two times of the bet if they *Surrender* after the first turn. If the player's hand exceeds 21 after *Hit* and *DoubleDown*, the player also loses immediately and forfeits the entire bet.

Once the player stops drawing, the dealer reveals the hole card and draws until the total count reaches at least 17. If the dealer's total count exceeds 21 or is lower than the player's total count, the player wins and gains an amount equal to the bet. If the player wins with an initial hand of A and a 10-valued card, they receive 1.5 times the bet. If two total counts are equal, then the player gains nothing. Otherwise, the player loses the bet.

**Input Structures.**    The cheater and the non-cheater receive observation consisting of five components: *Player's initial hand*, *Dealer's initial hand*, *Deck*, *Player's current count*, and *Player's action history*. Fig. 9 illustrates example game states with the observation. Each card in each component is encoded as a pair of its numeric value and a boolean flag that marks whether it is an A. Any card that is not currently visible to the player is masked with 0. After the player chooses *Hit* or *DoubleDown*, the top face-down card of the deck is revealed to the player. On the other hand, the cheater can observe every card, including the dealer's hole card and the entire deck. The cheat detector receives the same information as the cheater.

## B    EXPERIMENT SETTING DETAILS

We conducted all experiments on a machine with AMD EPYC 7742 (64-core), NVIDIA A100 (40GB) and 128GB of RAM. We use a CNN-based actor-critic architecture for the policy network and a CNN-based classifier for the detector network. Both the cheater and non-cheater policies are trained using Proximal Policy Optimization (PPO) algorithm (Schulman et al., 2017), with Generalized Advantage Estimation (GAE) (Schulman et al., 2015). We set the GAE parameter to 0.95, the clipping range to 0.2, and the value function loss coefficient to 0.5. We perform the training

|  | AP | | AUROC | |
|---|---|---|---|---|
|  | Detector (Pre.) | Detector (Adv.) | Detector (Pre.) | Detector (Adv.) |
| Pure cheater | 0.760 | 0.934 | 0.800 | 0.939 |
| Cheater (Adv.) | 0.733 | **0.905** | 0.774 | **0.911** |

Table 3: Pretrained (*Pre.*) and adversarially trained (*Adv.*) cheaters and detectors under $\lambda = 0.01$ in the Gridworld environment. Since the cheater's rewards are very similar, we ignore the reward component and only consider the detectability component when computing the equilibrium. Bold values indicate the equilibrium, where both cheaters and detectors are adversarially trained.

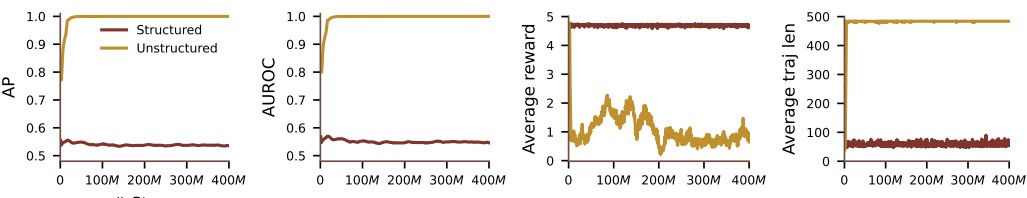

Figure 10: Effect of the structured cheater modeling in the Gridworld environment. We compare two settings under a fixed adversarial coefficient of $\lambda = 3.0$: one using the structured cheater modeling (*Structured*) and one using a standard, unstructured modeling (*Unstructured*). The unstructured modeling exhibits unstable reward patterns and inefficiently long trajectories throughout training. These results indicate that the structured modeling enables the cheater to learn more stably and effectively under strong adversarial pressure.

using 64 parallel environments, each generating rollouts of 2048 steps. PPO updates are applied over 4 epochs with a minibatch size of 1024. An entropy coefficient of 0.01 is used to encourage exploration. As for the detector, it is trained with a batch size of 8. We optimize both the policy and the detector networks using Adam optimizer (Kingma & Ba, 2014) with hyperparameters $\beta_1 = 0.9$, $\beta_2 = 0.999$, and a learning rate of $3 \times 10^{-4}$.

Before conducting adversarial training, we pretrain both the policy and detector networks. We pretrain the policies over one billion environment timesteps, and select checkpoints that maximize the reward. For the detector, we construct a trajectory dataset consisting of 10k training samples, 2k validation samples, and 2k test samples for each policy. We then pretrain the detector for 100 epochs and select the checkpoint with the lowest validation loss. The cheater and non-cheater policies were pretrained in about 2.5 days, and the detector was pretrained in roughly 10 minutes. The pretraining results are summarized in Tab. 1.

For adversarial training, we finetune the cheater policy and the detector network for an additional 400 million environment timesteps based on Alg. 1. The adversarial training phase took approximately 3 days. After training, we use the final checkpoints for performance evaluation.

To ensure the reliability of results, we conducted each experiment three times using different random seeds and reported the mean and the standard deviation.

## C STRUCTURED CHEATER MODELING STABILIZES TRAINING DYNAMICS

In this section, we compare the training dynamics of the cheater policies with the structured modeling and without employing the structured modeling. We design an unstructured modeling structure as only pure cheater network in Fig. 2. For the comparison, we set $\lambda = 3.0$ to highlight the stability issue that arises during training under strong adversarial pressure. As shown in Fig. 10, the structured modeling consistently maintains high average rewards and short, stable trajectories throughout training. In contrast, the unstructured modeling exhibits highly unstable reward patterns and significantly longer trajectories, indicating inefficient and erratic behavior. This erratic behavior makes the cheater's actions more distinguishable from those of non-cheaters, leading the detector to classify them with higher confidence. As a result, the detection performance for the unstructured modeling converges to near-perfect levels, with AP and AUROC approaching 1.0.

| | | Trajectory-length-based detector | | Reward-based detector | |
|---|---|---|---|---|---|
| Game | Player type | AP | AUROC | AP | AUROC |
| Gridworld | Pure cheater | $0.684_{\pm 0.002}$ | $0.730_{\pm 0.003}$ | $0.675_{\pm 0.000}$ | $0.702_{\pm 0.005}$ |
| Blackjack | Pure cheater | $0.752_{\pm 0.003}$ | $0.595_{\pm 0.011}$ | $0.717_{\pm 0.004}$ | $0.640_{\pm 0.005}$ |

Table 4: AP and AUROC of the pretrained agents with different detectors.

## D  EFFECT OF DETECTOR DESIGN

Feature engineering (i.e., selecting important features) plays a crucial role in cheat detection (Alayed et al., 2013). Detector performance can be changed significantly depending on the selected features. Previously, we used the trajectory as a feature to implement the detector. Alternatively, we could use different features such as trajectory length and reward.

In this section, we introduce two additional detectors: a trajectory-length-based detector and a reward-based detector. We design the detectors to reflect the fact that cheaters tend to have shorter trajectory lengths and larger rewards compared to non-cheaters. Therefore, the cheater probability of the trajectory-length-based detector $D_{len} : \mathbb{R} \to [0, 1]$ and the cheater probability of the reward-based detector $D_{rew} : \mathbb{R} \to [0, 1]$ can be defined as following logistic functions:

$$D_{len}(l) = \frac{1}{1 + e^{-(l-b_{len})/t_{len}}}, \quad D_{rew}(r) = \frac{1}{1 + e^{-(r-b_{rew})/t_{rew}}}, \tag{8}$$

where $l$ is a trajectory length, $r$ is a reward, $t_{len}$, $b_{len}$, $t_{rew}$ and $b_{rew}$ are learnable parameters.

Note that the trajectory length and the reward are not effective criteria for distinguishing cheaters from non-cheaters. It is because their values can vary widely depending on the objects' placement on the map or the values of the cards. As a result, detectors based on these measures achieve much lower AP and AUROC than the trajectory-based detector (Fig. 7a and 7b). Furthermore, cheaters can easily bypass the detectors due to their poor performance. In this case, cheaters can maintain low detectability while obtaining sufficiently high rewards even without considering the detector. Consequently, when the adversarial coefficient $\lambda$ is not large, metrics remain relatively stable in Fig. 11. In addition, it narrows the performance gap between training with and without the detector, as shown in Fig. 12, compared to what we observed with the trajectory-based detector in Fig. 5.

Except for these points, we observed a trend similar to that of the trajectory-based detector. As $\lambda$ increases, the cheater policy becomes more difficult to detect, resulting in lower AP and AUROC. To evade the detector, cheaters increase their trajectory length by taking longer and less direct routes. It also delays item collection in Gridworld or reduces the bet in Blackjack, leading to a lower reward.

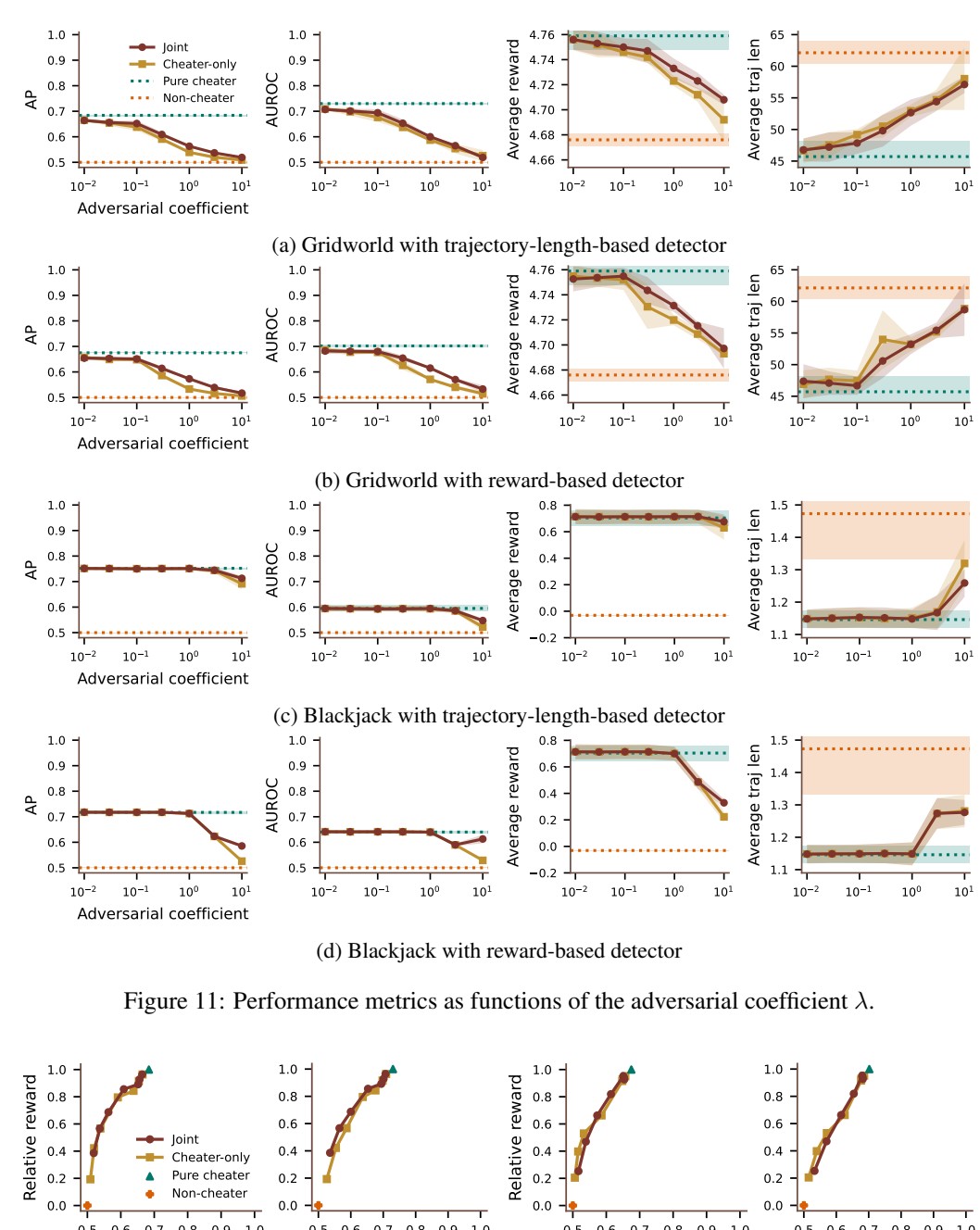

Figure 11: Performance metrics as functions of the adversarial coefficient $\lambda$.

(a) Gridworld with trajectory-length-based detector

(b) Gridworld with reward-based detector

(c) Blackjack with trajectory-length-based detector

(d) Blackjack with reward-based detector

(a) Gridworld with trajectory-length-based detector

(b) Gridworld with reward-based detector

(c) Blackjack with trajectory-length-based detector

(d) Blackjack with reward-based detector

Figure 12: Reward changes over detectability.

# E   THEORETICAL RELATIONSHIP BETWEEN THE CHEATER POLICY AND THE DETECTION PERFORMANCE

In this section, we study how the choice of the cheater policy affects the detection performance.

**Relationship between Cheater Policy and KL Divergence.**   KL divergence between the cheater policy $\pi_c$ and the non-cheater policy $\pi_n$:

$$
\begin{aligned}
D_{\mathrm{KL}}(\pi_c||\pi_n) &= \mathbb{E}_{(s,o)\sim\mathcal{S}\times\Omega}\mathbb{E}_{a_c\sim\pi_c}[-\log\pi_n(a_c|s,o)] - \mathbb{E}_{(s,o)\sim\mathcal{S}\times\Omega}H(\pi_c(\cdot|s,o)) \\
&\geq -\log\mathbb{E}_{(s,o)\sim\mathcal{S}\times\Omega}\mathbb{E}_{a_c\sim\pi_c}[\pi_n(a_c|s,o)] - \mathbb{E}_{(s,o)\sim\mathcal{S}\times\Omega}H(\pi_c(\cdot|s,o)),
\end{aligned}
\tag{9}
$$

where the inequality follows from Jensen's inequality. Let $q = \mathbb{E}_{(s,o)\sim\mathcal{S}\times\Omega}\mathbb{E}_{a_c\sim\pi_c}[\pi_n(a_c|s,o)]$ and $q^* = \mathbb{E}_{o\sim\Omega}[\max_{a\in\mathcal{A}}\pi_c(a|o)]$. Then,

$$
\begin{aligned}
q = \mathbb{E}_{(s,o)\sim\mathcal{S}\times\Omega}\mathbb{E}_{a_c\sim\pi_c}[\pi_n(a_c|s,o)] &= \mathbb{E}_{(s,o)\sim\mathcal{S}\times\Omega}\sum_{a\in\mathcal{A}}\pi_c(a|s,o)\pi_n(a|s,o) \\
&= \mathbb{E}_{(s,o)\sim\mathcal{S}\times\Omega}\sum_{a\in\mathcal{A}}\pi_c(a|s,o)\pi_n(a|o) \\
&= \mathbb{E}_{o\sim\Omega}[\mathbb{E}_{s\sim\mathcal{S}|o}[\sum_{a\in\mathcal{A}}\pi_c(a|s,o)\pi_n(a|o)|o]] \\
&= \mathbb{E}_{o\sim\Omega}[\sum_{a\in\mathcal{A}}\pi_n(a|o)\mathbb{E}_{s\sim\mathcal{S}|o}[\pi_c(a|s,o)|o]] \\
&= \mathbb{E}_{o\sim\Omega}[\sum_{a\in\mathcal{A}}\pi_n(a|o)\sum_{s\in\mathcal{S}}p(s|o)\pi_c(a|s,o)] \\
&= \mathbb{E}_{o\sim\Omega}[\sum_{a\in\mathcal{A}}\pi_n(a|o)\pi_c(a|o)] \\
&\leq \mathbb{E}_{o\sim\Omega}[\max_{a\in\mathcal{A}}\pi_c(a|o)] = q^*
\end{aligned}
\tag{10}
$$

holds. By Fano's inequality (Fano, 1952),

$$
1 - q^* \geq \frac{H(\mathcal{A}_c|\Omega) - \log 2}{\log|\mathcal{A}_c|} = \frac{H(\mathcal{A}_c|\Omega) - \log 2}{\log|\mathcal{A}|}
\tag{11}
$$

holds. Moreover, using the strong data-processing inequality (Pippenger, 2002), $H(\mathcal{A}_c|\Omega)$ has a following lower bound:

$$
\begin{aligned}
H(\mathcal{A}_c|\Omega) &= H(\mathcal{A}_c) - I(\mathcal{A}_c;\Omega) \\
&\geq H(\mathcal{A}_c) - \eta I(\mathcal{A}_c;\mathcal{S}\times\Omega) = (1-\eta)H(\mathcal{A}_c) + \eta H(\mathcal{A}_c|\mathcal{S}\times\Omega),
\end{aligned}
\tag{12}
$$

where $\eta \in [0,1]$ is a contraction coefficient. $\eta$ quantifies how much of the mutual information between $\mathcal{A}_c$ and $\mathcal{S}\times\Omega$ is preserved when passing through $\Omega$. A smaller $\eta$ indicates stronger information loss. Combining the above results, we obtain the following lower bound of $D_{\mathrm{KL}}(\pi_c||\pi_n)$:

$$
\begin{aligned}
D_{\mathrm{KL}}(\pi_c||\pi_n) &\geq -\log q - \mathbb{E}_{(s,o)\sim\mathcal{S}\times\Omega}H(\pi_c(\cdot|s,o)) \\
&\geq -\log q^* - \mathbb{E}_{(s,o)\sim\mathcal{S}\times\Omega}H(\pi_c(\cdot|s,o)) \\
&\geq -\log(1 - \frac{H(\mathcal{A}_c|\Omega) - \log 2}{\log|\mathcal{A}|}) - \mathbb{E}_{(s,o)\sim\mathcal{S}\times\Omega}H(\pi_c(\cdot|s,o)) \\
&\geq -\log(1 - \frac{(1-\eta)H(\mathcal{A}_c) + \eta H(\mathcal{A}_c|\mathcal{S}\times\Omega) - \log 2}{\log|\mathcal{A}|}) - H(\mathcal{A}_c|\mathcal{S}\times\Omega).
\end{aligned}
\tag{13}
$$

If $H(\mathcal{A}_c|\mathcal{S}\times\Omega)$ is sufficiently small (i.e., the cheater's action is nearly determined by $(s,o)$), the lower bound becomes positive. In addition, $H(\mathcal{A}_c|\mathcal{S}\times\Omega) = H(\mathcal{A}_c) - I(\mathcal{A}_c;\mathcal{S}\times\Omega) \leq H(\mathcal{A}_c)$ always holds. As a result, the lower bound of the KL divergence increases as $\eta$ decreases, which is likely to increase the KL divergence.

**Relationship between KL Divergence and Detection Performance.** Consider a binary hypothesis testing problem between $H_0 : \pi_c$ and $H_1 : \pi_n$ based on $m$ i.i.d. samples. By the Chernoff–Stein's Lemma (Thomas & Joy, 2006), the type-II error $\beta_m$ under a fixed and bounded type-I error $\alpha_m$ satisfies

$$\lim_{m \to \infty} -\tfrac{1}{m} \log \beta_m = D_{\mathrm{KL}}(\pi_c \| \pi_n). \tag{14}$$

Hence, a larger $D_{\mathrm{KL}}(\pi_c \| \pi_n)$ implies a lower asymptotic error, leading to improved detection performance.

**Conclusion.** In this setting, the contraction coefficient $\eta$ reflects the degree of information preservation from $\mathcal{S} \times \Omega$ to $\Omega$ under the given cheater policy. A smaller $\eta$ indicates greater information loss, which may arise from the cheater policy that heavily exploits the unobserved components of the state. Such a decrease in $\eta$ results in a larger lower bound of $D_{\mathrm{KL}}(\pi_c \| \pi_n)$, and thus higher detection performance. To summarize,

$$\text{Greater information loss} \ \Rightarrow \ \text{Larger lower bound of } D_{\mathrm{KL}}(\pi_c \| \pi_n)$$
$$\Rightarrow \ \text{Likely to result in larger } D_{\mathrm{KL}}(\pi_c \| \pi_n) \tag{15}$$
$$\Rightarrow \ \text{Higher detection performance.}$$

