# OpenReview forum: "Adversarial Reinforcement Learning Framework for ESP Cheater Simulation"
_ICLR.cc/2026/Conference — Submitted to ICLR 2026_

### Official Review · Reviewer_t5MH · 2025-10-20

**Soundness:** 3
**Presentation:** 3
**Contribution:** 3
**Rating:** 8
**Confidence:** 3

**Summary:**

The paper presents an approach/framework for the study of adaptive cheaters and the development of efficient cheat detectors in games. A simulation approach is adopted, with cheaters and non-cheaters modelled as RL agents and the cheat detector as a classifier, with the interactions modelled as an adversarial game. Experiments with two simple environments are conducted to demonstrate the framework and the results analysed.

**Strengths:**

This is a well written paper. The methodology is clearly laid out and well presented.

The experiments are comprehensive and thorough, and the analysis is considered and sound. The conclusion that the detector developed with adversarial training is a more robust classifier is a solid result, as well as the observation that the adversarially trained cheater can evade detection while still enjoying improved rewards over a non-cheater.

Though not a core area for me, this seems to be a solid contribution, with some clearly elucidated directions for future research too.

**Weaknesses:**

No weaknesses to note.

Minor: not sure AUROC was defined.

**Questions:**

Trivial point - for ease of reference, the baseline numbers from Table 1 might have been good to add as bars in Figure 4, so as to be able to get a better sense of the behaviour as lambda varies (and to more clearly see the excess AP performance for small lambda)

---

> ### Author Response · Authors · 2025-11-23
>
> Thank you for the thoughtful comment. We have reflected your suggestions in the revised paper.
>
> `W1. Definition of AUROC`
>
> Thank you for pointing this out. AUROC (Area Under the Receiver Operating Characteristic curve) measures the area under the ROC curve that represents the trade-off between true and false positive rates. A higher AUROC indicates a stronger ability to identify cheaters and avoid false alarms. We’ve clarified the definition and the explanation of AUROC in the revised paper (Section 4.1, Experiment Setup, Line 361-363).
>
> `Q1. Adding baseline numbers in Figure 4. `
>
> Thank you for the helpful suggestion. Following your comment, we’ve updated Figures 4 and 11 in the revised paper to include the baseline results (pure cheater, non-cheater) as bars.

---

> > ### Comment · Reviewer_t5MH · 2025-11-24
> >
> > Thank you

---

### Official Review · Reviewer_E6T1 · 2025-10-30

**Soundness:** 2
**Presentation:** 3
**Contribution:** 1
**Rating:** 4
**Confidence:** 4

**Summary:**

This paper proposes an adversarial framework to model cheaters and anti-cheater detection in computer games. It fuses standard RL with a minmax objective to train both models at the same time. Experimental analysis is done on two toy environments and several configurations are studied, including sensible baselines based on reward and episode length.

**Strengths:**

This paper proposes a relatively realistic model of cheating in games, which supports the cheater's dual objectives of maximizing reward while avoiding detection, and the mutual co-adaptation of cheater and detector. While it is not ground-breaking (being based on fairly well-established adversarial frameworks), it seems well-conceived.

The paper itself is clear, with only a few areas that could use improvement.

The experiments, while done in small settings, do focus on the main areas of interest: adaptive vs non-adaptive behaviour, comparing with proxy signals such as success and episode length, and sensitivity analysis of the coefficient trading off detection vs reward maximization.

**Weaknesses:**

The main weakness of the paper is in the experimental validation. While the analysis is sound, the conclusions are limited because only two very simple settings were studied. An analysis of more realistic (if small) games would be more conclusive.

The lack of such a comparison is probably explained by another weakness of the methodology: it requires white-box access to RL policies of cheaters and non-cheaters. This means that it cannot be easily adapted to existing games, where game developers can collect datasets of player behaviour, and it is not obvious which are cheaters; but due to the complexity of modern games, it is not a trivial endeavour to train effective RL policies for them.

Another aspect that is understudied is variability in player skill. Both cheater and non-cheater are assumed to play as well as possible (given the limits of training). It is relevant to study the spectrum of non-cheater skill -- especially as high-skill players may be labelled as cheaters, depending on the game. Cheater skill level is also relevant, as a player with poor skills who uses cheating may be easier to detect than a skilled one.

The related work section is extremely short, making it seem like there are more works that were left out. As just one example, Franzmeyer et al. "Illusory Attacks" (ICLR 2024) also seems to study attackers that attempt to conceal their interference, but the frameworks may be different (maximize a player's reward vs minimize a victim's reward).

Fig. 1: The difference between cheater and non-cheater is unclear - would be better to have them interact with a single environment, and having an extra arrow from the environment that represents unfair (ESP) observations. The feedback going to both cheater and detector from itself is also a confusing way to present it.

**Questions:**

I would like the authors to comment on the relevance of their contribution in the context of the difficulty of training good RL agents in realistic settings, and how it affects extensions to commercial games.

---

> ### Author Response · Authors · 2025-11-23
> **Official Comment by Authors (1/2)**
>
> We thank the reviewer for the constructive feedback. We have reflected your feedback in the revised paper.
>
> `W1. An analysis of more realistic (if small) games would be more conclusive.`
>
> We initially explored more realistic video games such as Doom and Atari. However, we concluded that the Gridworld and Blackjack environments would be the most efficient choice for our experiment when taking the following practical constraints into account.
>
> 1. The agent must be trainable by reinforcement learning to achieve sufficient performance.
> 2. The environment must also support both fully observable and partially observable agents.
> 3. The detector must accurately distinguish cheaters from non-cheaters. Otherwise, cheaters could get high rewards while remaining undetected even without adversarial training.
> 4. Training must be completed within a reasonable timeframe, because adversarial training requires many iterations for the detector and cheater to reach equilibrium.
>
> Our environments meet all of these requirements. In particular, they render every object and trajectory in a single image, allowing us to use a simple image-based classifier as our detector, and speed up training. In contrast, environments like Atari require a video-based classifier, which significantly slows down the training process.
> We acknowledge that our environments are much simpler than real-world games. As we mentioned in Section 5, Conclusions and future works, we will expand our experiments to more realistic scenarios in future work.
>
> `W2. Due to the complexity of modern games, it is not a trivial endeavour to train effective RL policies for them.`
>
> This limitation is an inherent risk of simulation-based approaches. Below are the pros and cons of the simulation and real-world dataset.
>
> [Simulation dataset]
> - Pros. We can fully control the behaviors of cheaters and non-cheaters. It allows us to generate large-scale trajectory data with accurate ground-truth labels.
> - Cons. It requires training RL agents. Complex real-world games would require implementation costs.
>
> [Real-world dataset]
> - Pros. It is possible to collect player behavior data directly during live game service.
> - Cons. Datasets are prone to distortion due to mislabeling, imbalance, and intentional deception by cheaters. Such issues make it difficult to obtain reliable data or construct sufficiently large labeled datasets.
>
> While simulation-based approaches may be more challenging to implement, they provide a controllable setting that real-world data cannot offer. As discussed in the introduction, this controllability allows us to simulate the cheater’s continuously evolving strategies and to generate large-scale datasets with accurate ground-truth labels. It is critical for analyzing adaptive behavior and validating detectors.
>
> `W3. Player skill variability is understudied.`
>
> To ensure a clear understanding and thorough analysis, we make a strong assumption that both the cheater and the non-cheater are using local optimal policies. This assumption can produce a large performance gap between the two agents, making it easier for the detector to distinguish them than in real-world scenarios. In other words, this setup represents an upper bound on the detector’s performance.
>
> Although the setup is disadvantageous to the cheater, they still act strategically, achieving high rewards while avoiding detection. We demonstrated that it is possible to adjust the balance between reward and detectability. In real-world scenarios where a range of cheaters exist and detector performance may be weaker, this suggests that cheaters could gain even greater benefits. To create more realistic simulations, we plan to explore a variety of cheating and non-cheating pairings, as we mentioned in our future work.
>
> We have clarified this point in the revised paper (Section 3, ESP Simulation Framework (Line 205-208); Section 5, Conclusions (Line 513-520))
>
> `W4. The related work section is extremely short.`
>
> Thank you for pointing this out. As suggested, we’ve expanded the Related Work section to include additional studies on adversarial attacks, including "Illusory Attacks". (Section 2, Related works, Line 135-173)

---

> ### Author Response · Authors · 2025-11-23
> **Official Comment by Authors (2/2)**
>
> `W5. Fig. 1: The difference between cheater and non-cheater is unclear - would be better to have them interact with a single environment, and having an extra arrow from the environment that represents unfair (ESP) observations. The feedback going to both cheater and detector from itself is also a confusing way to present it.`
>
> If the figure shows the cheater and the non-cheater interacting with a single environment, readers might misinterpret the framework as a multi-agent setting. Since this paper focuses on the single-agent environment, we would like to separate the environments to prevent any potential misunderstanding.
>
> Next, we replace the term 'Feedback' with more specific descriptions to clarify its meaning and improve readability; 'Cheater probability as negative reward' and 'model update via gradient descent'.
>
> `Q1. Relevance of their contribution in the context of the difficulty of training good RL agents in realistic settings, and how it affects extensions to commercial games.`
>
> We acknowledge that training RL agents in realistic game environments is challenging due to the complexity of the state space, computational cost, and engineering effort required.
> Recently, however, many commercial game studios have begun training RL agents within their game environments for in-game bots or QA testing, such as Ubisoft [a], Sony [b], Tencent [c], and Electronic Arts (EA) [d]. From this perspective, the use of RL agents in real game environments is both feasible and practical.
>
> Moreover, in commercial games, labeling player data with precise cheat annotations typically requires extensive manual effort by human reviewers. In particular, for ESP cheats, even human reviewers struggle to determine whether a player is cheating based solely on gameplay logs, which can lead to low labeling quality.
> Instead of relying on such imperfect data, using precisely labeled data, as in our framework, enables accurate evaluation of key performance metrics such as accuracy and false-positive/false-negative rates. It enhances the reliability of detection performance analysis and improvement.
>
> We therefore believe that our framework provides a practical foundation that could complement existing data-driven approaches used in real-world anti-cheat systems by game companies.
>
> ---
> [a] Istander et al., Reinforcement Learning Agents for Ubisoft’s Roller Champions, NeurIPS 2020 Challenges of Real-World RL workshop.
>
> [b] Wurman et al., Outracing champion Gran Turismo drivers with deep reinforcement learning, Nature 2022.
>
> [c] Wei et al., Honor of Kings Arena: an Environment for Generalization in Competitive Reinforcement Learning, NeurIPS 2022.
>
> [d] Gillberg et al., Technical Challenges of Deploying Reinforcement Learning Agents for Game Testing in AAA Games, CIG 2023.

---

> ### Author Response · Authors · 2025-11-27
> **Gentle reminder**
>
> Dear Reviewer,
>
> Thank you once again for your time and thoughtful review of our paper.
>
> As the discussion period is about to conclude, we would like to remind you that if you have any remaining comments, please share them.
>
> We believe we have sincerely and successfully addressed your concerns, supported by the additional results and the carefully revised draft.
>
> If you have any further concerns or questions, please feel free to let us know.
>
> Thanks a lot,
>
> Authors

---

> > ### Comment · Reviewer_E6T1 · 2025-11-27
> >
> > Thank you for your responses. I think that the main outstanding issue is the unrealistic settings that were chosen, which are understandable because of the limitations of the framework; but then again, these limitations do point to a fragility of the framework itself. It may be the case that a different paradigm than adversarial RL training is needed to allow scaling up. If not, then the authors need to demonstrate that scaling in practice. I would be ready to excuse the small scale for a theory paper, but this is not the case.
> >
> > Atari is not adequate also because there is hardly any hidden information. But I don't believe that RGB inputs are needed. The RL agents used in commercial games don't use RGB, but rather a compact description of the (visible) game state, for speed.
> >
> > It's not readily apparent why the authors did not do this. It would be possible to do it for Doom (which is open-source and very hackable, e.g. https://github.com/vlfeat/researchdoom-chocolate), for Quake if multi-player arenas are needed, and even Starcraft (Google DeepMind made available a research API that gives full access to the game state).
> >
> > In all cases, limited views (for example of adversaries) can be simulated by raycasts, and the game info passed to a MLP as a fixed-size vector. All in all, I don't think that the authors realized the full potential of the research that they are proposing.

---

### Official Review · Reviewer_ZEf6 · 2025-10-31

**Soundness:** 3
**Presentation:** 3
**Contribution:** 2
**Rating:** 4
**Confidence:** 3

**Summary:**

This paper constructs a simulation methodology that allows for reproducing ESP cheater behavior through adversarial reinforcement learning, including an adaptive cheat detector and adversarial training of cheaters.
The simulation combines existing techniques adapted to this problem setting, including environment formulation via POMDP/MDP, GAN/GAIL-style min-max training, reward shaping, PPO optimization, and trajectory classifier detection. Based on experiments, this study argues that detecting such adaptive cheaters requires a co-evolving detector.

**Strengths:**

- This research targets a realistic and socially significant problem, ESP cheating, that has not been intorduced by conventional adversarial RL or security RL approaches.
- The mathematical formulation of the proposed framework is well-organized.

**Weaknesses:**

- The framework of ESP cheaters versus detectors appears to be a new application, but it is essentially a form of adaptive RL-based evasion and bears strong similarities to existing RL-based adversarial attacks against detectors/classifiers.
- To the best of the reviewers' knowledge, the novelty in the design lies in:
(1) the detector being based on trajectory-based cheetah detection, and (2) introducing a complementary structure for cheetahs similar to MoE or a gating network. Both represent straightforward applications of existing techniques to this problem setting.

**Questions:**

The difference in observation where non-cheaters follow a POMDP while cheaters follow an MDP is critically important concerning the nature of the problem. How does this observational difference affect the difficulty of detection or convergence to equilibrium? To what extent does the information gap between partial observation and full observation bring about qualitative changes in the structure of the game? Can theoretical or empirical insights be provided on this point?

---

> ### Author Response · Authors · 2025-11-23
>
> We thank the reviewer for the constructive feedback. We have reflected your feedback in the revised paper.
>
> `W1. The proposed framework is similar to existing RL-based adversarial attacks against detectors/classifiers.`
>
> RL-based adversarial attack focused on improving the robustness of reinforcement learning agents or developing attack methodologies that effectively disrupt a given agent. To the best of our knowledge, most adversarial attacks directly target the observation or policy. In contrast, the goal of this paper is not to design a more robust cheater or a superior detector, but to model and analyze the interactive co-adaptation process in which the cheater and the detector continuously respond to each other’s policy/classifier changes. Furthermore, unlike prior approaches, we do not manipulate the observation or policy directly. Instead, the detection probability is provided as a negative reward, allowing the cheater to learn adaptive behaviors through this indirect feedback.
>
> `W2. The design novelty is limited to using trajectory-based detection and a MoE-like complementary structure. Both are straightforward applications of existing techniques to this problem setting.`
>
> As the reviewer mentioned, our structured cheater modeling is conceptually inspired by MoE architecture. Accordingly, we have added the relevant MoE papers as additional references in the revised paper (Line 262-265).
>
> However, to the best of our knowledge, no prior work in the cheat detection domain has modeled the adaptive behavior of ESP cheaters. While the idea of a gating mechanism has been explored in other domains, this is the first work to formulate and apply such a structure to simulate the adaptive dynamics of cheaters.
>
> `Q1. How does the POMDP vs. MDP observation difference affect the difficulty of detection or convergence to equilibrium?`
>
> In Appendix E of the revised paper, we’ve included a theoretical analysis linking the cheater’s observability to the detector’s performance. In summary, we show that as the information loss from the state-observation space ($S\times\Omega$) to the partial observation space ($\Omega$) increases, the lower bound of the KL divergence between the cheater and the non-cheater policies also increases. This results in a larger KL divergence and consequently enhances the detection performance. As a result, greater observability makes the cheater's behavior more distinguishable from that of the non-cheater, allowing the detector to differentiate them more easily.
>
> `Q2. To what extent does the information gap between partial observation and full observation bring about qualitative changes in the structure of the game?`
>
> When the information loss from $S\times\Omega$ to $\Omega$ is large, the detector can more easily capture the differences in behavioral patterns between the two groups, resulting in higher detection performance. To reduce the likelihood of being detected, the cheater intentionally restricts the use of available information, thereby shaping a behavioral distribution that is closer to that of the non-cheater.
>
> In our framework, the simulated cheaters learn the trade-off between the non-cheater policy and the pure-cheater policy, intentionally excluding certain information to minimize detection risk. We can observe this phenomenon in Figure 6, where the degree of this trade-off correlates with gradual qualitative changes in the trajectory patterns.

---

> ### Author Response · Authors · 2025-11-27
> **Gentle reminder**
>
> Dear Reviewer,
>
> Thank you once again for your time and thoughtful review of our paper.
>
> As the discussion period is about to conclude, we would like to remind you that if you have any remaining comments, please share them.
>
> We believe we have sincerely and successfully addressed your concerns, supported by the additional results and the carefully revised draft.
>
> If you have any further concerns or questions, please feel free to let us know.
>
> Thanks a lot,
>
> Authors

---

### Official Review · Reviewer_tTX8 · 2025-11-01

**Soundness:** 2
**Presentation:** 3
**Contribution:** 2
**Rating:** 2
**Confidence:** 3

**Summary:**

The paper proposes a simulation framework to model the adversarial interaction between an ESP cheater and a cheat detector. The interaction between an adaptive cheater and a detector is then formulated as a minimax game. The framework is tested on custom Gridworld and Blackjack environments.

**Strengths:**

Novel Architectural Choice: The Structured Cheater Model is a clever and novel contribution. By freezing the base policies and only learning the interpolation function $\omega$, the authors simplify the adversarial learning problem.

Well-Motivated Problem: The paper addresses a practical and difficult problem (detecting adaptive cheaters) where collecting labeled data is notoriously hard. The simulation-based approach is well-justified .

**Weaknesses:**

Disconnect Between Motivation and Experiments: This is the most significant weakness. The paper is motivated by complex, multiplayer First-Person Shooter (FPS) games where ESP cheats are a major issue. However, the experiments are conducted on simple, single-agent Gridworld and Blackjack environments. These environments fail to capture the strategic, interactive, and high-dimensional nature of the motivating problem. The authors acknowledge this as future work, but the gap is too large for the claims to be considered general.

Limited Novelty of the Base Framework: While the structured model is novel, the underlying adversarial RL framework (a minimax game solved with GDA and a non-saturating generator loss ) is a standard formulation, similar in principle to GANs and Generative Adversarial Imitation Learning (GAIL). The paper's novelty rests almost entirely on the structured architecture rather than a new learning paradigm

**Questions:**

Please see the weakness

---

> ### Author Response · Authors · 2025-11-23
>
> Thank you for the constructive comments. We have reflected your feedback in the revised paper.
>
> `W1. Large gap between motivation (complex multiplayer FPS) and experiments (simple, single-agent Gridworld and Blackjack environment)`
>
> As the reviewer pointed out, complex environments such as multiplayer FPS games are the long-term goal of our research. However, these environments are highly non-stationary due to the player interactions and the dynamic behaviors of opponents, which makes it difficult to clearly interpret the specific factors that drive the adaptive changes of cheaters and detectors.
>
> To first understand these dynamics in a controllable and interpretable setting, we employed simplified single-agent environments that provide well-defined behavioral indicators, such as exploration and trajectory length. Although these environments are abstracted, they capture fundamental aspects such as spatial exploration and reward-driven behavior that also appear in more complex game settings. In particular, even in FPS games, player movements and interactions can often be represented as grid-like trajectories on a minimap, making the Gridworld environment a meaningful abstraction for studying such dynamics.
>
> Consequently, this paper focuses on establishing a well-defined baseline in a single-agent setting and investigating the co-adaptation mechanism between cheaters and detectors under controlled conditions, before extending the analysis to complex multi-agent environments.
> We have clarified this point in the revised paper (Section 1, Introduction; Line 99-109)
>
> `W2. Limited Novelty: The adversarial RL framework is a standard formulation like GAN or GAIL. The novelty rests only on the structured architecture.`
>
> The main contribution of this work is to establish a simulation framework that can model adaptive cheating behaviors under information asymmetry. Although the overall formulation is similar to the minimax structure of GAN or GAIL, the modeling purpose is fundamentally different.
>
> GAN and GAIL aim to imitate the data distribution or policies, focusing on designing highly expressive generators and improving distribution-matching performance. In contrast, the cheater in our framework does not necessarily imitate the non-cheater’s policy. It instead learns new behaviors that balance the trade-off between detection avoidance and reward maximization. While imitation may help evade detection, it cannot achieve the cheater’s primary goal, which is obtaining unfair advantages through higher rewards. As shown in Figure 4, we analyze how the behaviors of both the cheater and the detector behave under different values of the adversarial coefficient \lambda.
>
> Another key difference of our study is that we treat the detector’s performance as an equally important subject of analysis. In conventional GAN or GAIL frameworks, the discriminator serves merely as an auxiliary tool for generator training, and its own performance is rarely analyzed. In contrast, our framework considers the detector as an active player. As shown in Figure 5, we evaluate how the detector’s performance changes in response to the cheater’s strategic behavior. This approach reflects the unique characteristics of the cheat detection domain and provides a clear conceptual distinction from GAN and GAIL.
>
> We have clarified this point in the revised paper (Section 2, Related Works; Line 162-173)

---

> ### Author Response · Authors · 2025-11-27
> **gentle reminder**
>
> Dear Reviewer,
>
> Thank you once again for your time and thoughtful review of our paper.
>
> As the discussion period is about to conclude, we would like to remind you that if you have any remaining comments, please share them.
>
> We believe we have sincerely and successfully addressed your concerns, supported by the additional results and the carefully revised draft.
>
> If you have any further concerns or questions, please feel free to let us know.
>
> Thanks a lot,
>
> Authors

---

### Meta-Review · Area_Chair_xovX · 2025-12-04

**Summary:**

The main critiques raised across the reviews center on the experimental scope, particularly the gap between the motivating real-world setting (complex, high-dimensional games with partial observability) and the simplified single-agent environments used in the current experiments. Several reviewers also noted that the methodological novelty lies more in the application and architectural structuring than in introducing a new theoretical or algorithmic paradigm.

My own primary concern aligns most closely with Reviewer E6T1. While I understand the practical constraints and appreciate the rationale the authors give for using tractable environments, I believe that demonstrating at least one experiment in a more complex environment would significantly strengthen the paper’s claims. There are many feasible ways to simulate partial observability or “hidden information” even in high-dimensional settings using synthetic manipulations. Pretend that the RGB signal (in DOOM and other FPS games) is the full state of the agent which is passed on to the cheater unaltered. Then construct non-cheater observations by applying controllable transformations, e.g., Gaussian blurring that increases with distance from a center pixel, masking, occlusion fields, or restricted-radius vision while leaving the cheater’s observations unaltered. This approach would allow the authors to emulate realistic information asymmetry. Even a single experiment of this nature in a more complex environment (e.g., Doom, Quake, or a grid-based rendering of a larger 3D scene) would have made the empirical validation feel substantially more complete. Moreover, in such a setup it would also be nice, potentially in the synthetic experiments too, to show evidence on the performance of the detector as you vary the observation space in favor for the cheater.

I strongly encourage the authors to continue developing this line of research and, in future iterations, incorporate more realistic simulation environments or synthetically manipulated partial-observation pipelines. With that addition, the work has the potential to make a significant impact. As it stands now, I find it hard to accept the paper only purely based on the synthetic games and hence my decision.

**Reviewer Concerns:**

see above

**Reviewer Scores:**

see above

---

### Decision · Program_Chairs · 2026-01-26

Reject